# The importance of making testable predictions: A cautionary tale

**Emma S. Choi, Erik Saberski[iD], Tom Lorimer, Cameron Smith, Unduwap Kandage-don[iD], Ronald S. Burton, George Sugihara***

Scripps Institution of Oceanography, University of California San Diego, La Jolla, CA, United States of America

\* gsugihara@ucsd.edu

**Data Availability Statement:** Data are available on GitHub (https://github.com/SugiharaLab/Projects/tree/master/Spring%20Temperature%20Triggers).

## Abstract

We found a startling correlation (Pearson ρ > 0.97) between a single event in daily sea surface temperatures each spring, and peak fish egg abundance measurements the following summer, in 7 years of approximately weekly fish egg abundance data collected at Scripps Pier in La Jolla California. Even more surprising was that this event-based result persisted despite the large and variable number of fish species involved (up to 46), and the large and variable time interval between trigger and response (up to ~3 months). To mitigate potential over-fitting, we made an out-of-sample prediction beyond the publication process for the peak summer egg abundance observed at Scripps Pier in 2020 (available on bioRxiv). During peer-review, the prediction failed, and while it would be tempting to explain this away as a result of the record-breaking toxic algal bloom that occurred during the spring (9x higher concentration of dinoflagellates than ever previously recorded), a re-examination of our methodology revealed a potential source of over-fitting that had not been evaluated for robustness. This cautionary tale highlights the importance of testable true out-of-sample predictions of future values that cannot (even accidentally) be used in model fitting, and that can therefore catch model assumptions that may otherwise escape notice. We believe that this example can benefit the current push towards ecology as a predictive science and support the notion that predictions should live and die in the public domain, along with the models that made them.

## Introduction

To comprehend the population dynamics underpinning biodiversity and essential ecosystem services, a heavy emphasis is placed on driving mechanisms, both biotic and abiotic. In marine environments, where fish stocks are of substantial ecological and economic interest, drivers need to be untangled to inform effective, practical, and sustainable environmental policy. Temperature is a particularly important (and sometimes controversial [1–3]) driver for fish and other marine ectotherm populations. Here we focus on the well-studied relationship between temperature and fish reproduction [4–6].

At the seasonal timescale, trends between water temperature and spawning activity have been observed in many fish species [7–16]. Huber and Bengtson [11] found that the gonads of

**Funding:** This work was supported by DoD-Strategic Environmental Research and Development Program 15 RC-2509 (GS), NSF DEB-1655203 (GS), NSF ABI-1667584 (GS), DOI USDI-NPS P20AC00527 (GS), the Scripps Institution of Oceanography Postdoctoral Fellowship (TL), the McQuown Fund and the McQuown Chair in Natural Sciences, University of California, San Diego (GS). Fish egg collection and identification was supported in part by the Richard Grand Foundation and the California Ocean Protection Council R/OPCSFAQ-12 (RB).

**Competing interests:** The authors have declared that no competing interests exist.

inland silversides (*Menidia beryllina*), a summer spawning species, did not mature to a reproductive level in the absence of increasing water temperatures. In yellow perch (*Perca flavenscens*) both decreasing autumn temperatures and low winter temperatures have been deemed critical in order for gonadal maturation to occur [12, 13]. Additionally, these fish have been manipulated to spawn earlier in the year by increasing the rate of water temperature change [13]. Kayes and Calbert [14] found that for the same yellow perch species, increasing temperature heightened egg production, but even in the absence of a temperature cue endogenous factors could induce spawning. In cyprinid fishes, the initiation of gametogenesis requires low temperatures, but the completion of the process requires increasing temperatures [7]. Other notable studies use degree days, a measure of time based on temperature, to track gonad development from the initiation of vitellogenesis to the onset of spawning [17, 18]. A study done by Henderson et al. [19] demonstrates that the timing of spring transitions and the duration of summer, defined by a temperature threshold, is related to shifts in the center of biomass for multiple species during their seasonal migration to spawning grounds, however, the shifts observed differ by species. These and other varying and apparently complicated specific effects suggest that more general quantitative relationships covering diverse species may be hard to come by.

Despite this, recently a strong quantitative predictive relationship was detected between average winter temperature and average spring-summer egg abundance for a suite of near-shore-spawning species off the coast of southern California [15], which was subsequently supported by out-of-sample data acquired the following year [16]. This relationship is largely explained by colder waters being indicative of large scale upwelling; a process known to supply nutrients to shallower waters [20].

Building on this encouraging result, we re-examined the data of [15] and [16], but now including the additional 2019 and 2020 data that have since become available, and found a true out-of-sample confirmation of that relationship (Fig 1C). To emphasize, that means the original relationship of Ref. [15] which was based on just five years of available data, has reasonably predicted the subsequent three years of egg abundance. The data underlying this relationship, which consist of approximately weekly-sampled, species-identified egg counts of 46 near-shore-spawning species from Scripps Pier since 2013 (see Methods) also contain substantial, and possibly important, fine-timescale information that was not considered in the seasonal relationship described in [15]. Though the statistical seasonal association is compelling, it emerges only in large-scale averages. By coupling the full-resolution fish egg abundance time series (Fig 1A) with daily-averaged sea-surface temperatures (Fig 1B) from the Southern Californian Coastal Observational Ocean Monitoring System (SCCOOS) dataset, we asked whether finer-timescale temperature dynamics provide information about finer-timescale fish egg abundance dynamics.

## Results

The high time-resolution data (Fig 1A and 1B) did not show a linear cross-correlation between the daily spring temperature and lagged daily egg abundance, with only weak relationships across all delays at this fine daily timescale (Fig 1D). However, in accordance with previous work [23–26], the S-Map test for nonlinearity [21] revealed that the egg abundance is driven by nonlinear processes (forecasts improve as the nonlinear parameter, $\theta$, is increased, Fig 1E). Further, convergent cross-mapping, a tool for detecting nonlinear coupling in dynamical systems [22] suggested that temperature has a nonlinear effect on egg abundance (converges to $\rho$ = 0.58, n = 295, Fig 1F). Thus, we expected that a daily timescale relationship may be detectable, just not with linear correlation.

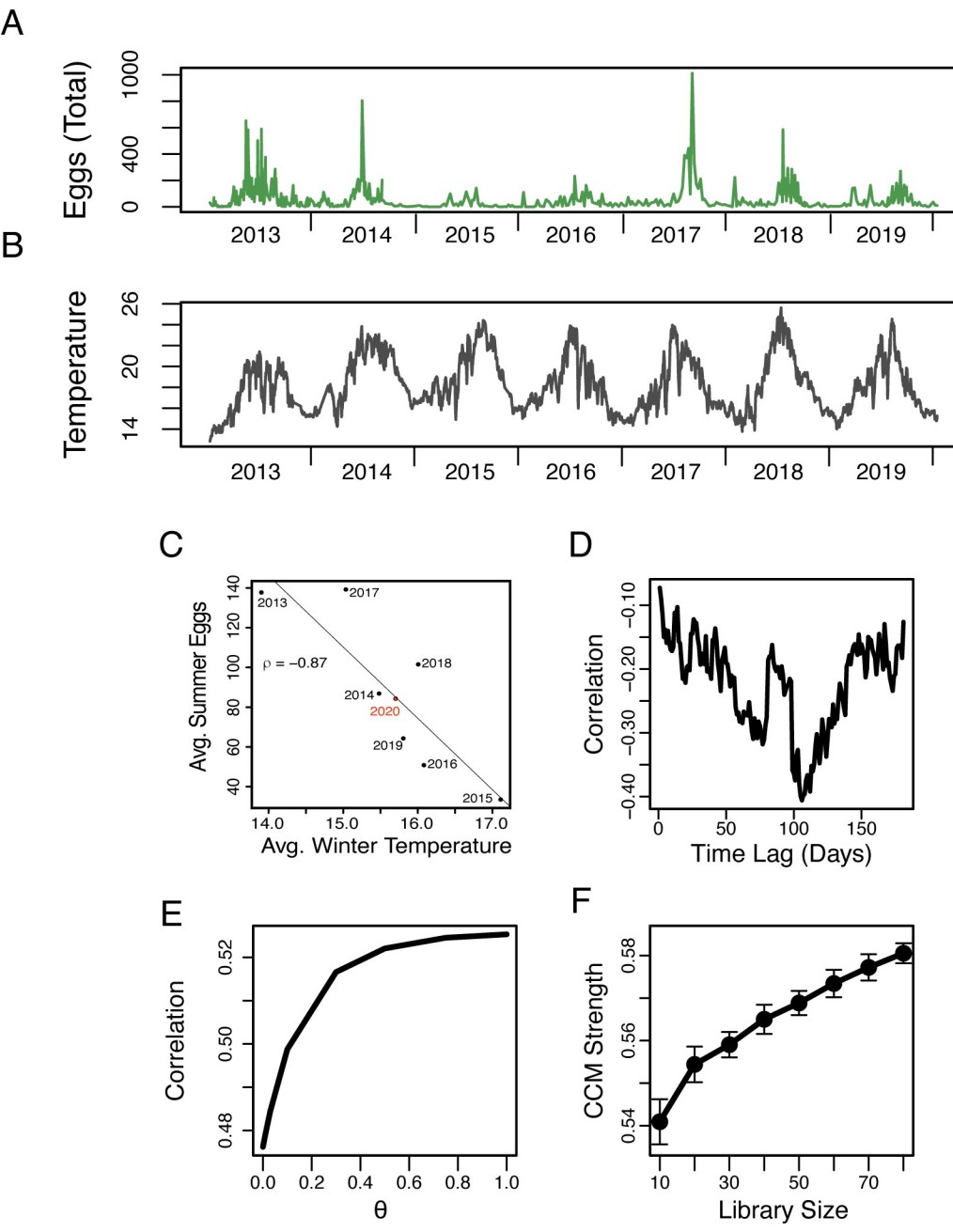

**Fig 1. Is there a fine-time-scale relationship between temperature and eggs?** A) The total egg abundance in each collection (see Methods) shows substantial variability from year to year in both mean and peak levels. Each fish egg collection was made from the Scripps Institution of Oceanography (SIO) Pier. B) The daily averaged sea surface temperature (SST) in ˚C at the SIO Pier from data taken every 5–10 minutes from the SCCOOS monitoring station. C) Seasonal averaging reveals the strong negative correlation between the average winter (December–February) SST and the average spring and summer (March–August) egg abundance, identified by [15], with additional points for 2018 [16], and now 2019 and 2020. D) The seasonal correlation breaks down at the daily level; there is no similarly strong correlation between daily winter temperatures and daily egg abundances with time delays ranging from 0 to 180 days. E) The S-Map [21] test for nonlinearity shows that forecasts of egg abundance improve (correlation between predictions and observations) as the nonlinearity parameter (θ) is increased, indicating that egg abundance shows nonlinear behavior. F) Convergent cross-mapping (CCM, [22]), shows that when using the egg abundance time series to map onto the temperature time series, predictions improve as library size increases, indicating there is a dynamic causal effect of daily-averaged temperature on egg abundance.

One type of event that stands out in the egg abundance time series (Fig 1A) is the peak in summer egg abundance. Both the magnitude and timing of the peak egg abundance varies from year to year with no obvious pattern. Previous studies indicate that increasing water temperature may provide a cue for spring and summer spawning species [19, 27–29]. To ascertain whether a relationship exists between spring temperature increase and peak summer egg abundance, we defined a generic *spring temperature trigger* (STT). Our STT is the maximum of all temperature increases detected within a moving window of length L, as that window moves over the spring season (Fig 2A, see Methods). This returns a single scalar value for the season, corresponding to a single event with an interpretable characteristic timescale (L). We restricted our analysis of temperature to the spring season (i.e. the season preceding the summer peak) following roughly the causal timescale examined in [15]. By examining a range of possible window lengths, we found a robust relationship around the 1 month timescale, between STT and peak summer egg abundance (L between 3 and 5 weeks, $\rho$ > 0.95; Fig 2B). This relationship was so remarkably strong ($\rho$ up to 0.98; Fig 2C), and apparently robust (Fig 2B) that we felt compelled to share this observation, despite the small number of data points involved (n = 7). To mitigate potential overfitting, we offered a prediction for the 2020 peak summer egg abundance that at the time of writing had not yet been measured (Fig 2C) [30]. To examine whether this relationship was caused by a general spring warming trend, we repeated the analysis on increasingly smoothed (time averaged) temperature data. We found that the predictive relationship from STT to peak summer egg abundance decreased markedly as temperature data became increasingly smoothed (Fig 3), which suggested to us that the information was indeed contained in the daily-resolution temperature information, and not in the trend.

The failure of our published out-of-sample prediction for 2020 [30] naturally led us to ask whether an exogenous change in conditions had come into play. Indeed, 2020 has been an anomalous year in many ways, and for marine life at Scripps Pier it was most notably seen in a toxic algal bloom (red tide) that was record-breaking both in terms of density of dinoflagellates (9x higher than ever previously recorded) and duration (over a month from early April until mid-May compared to typical 1–2 week blooms see https://sccoos.org/california-hab-bulletin/red-tide). This led not only to extraordinary fish kills (including within experimental and educational aquaria associated with Scripps that became contaminated by seawater intake) but also modified other physical (e.g. optical) and chemical properties of the near-shore marine environment, with a broad impact across that ecosystem. In many respects, it would have been surprising if fish spawning in 2020 was *unaffected* by this red tide. Moreover, the direction of the error in our prediction was consistent with these changes. The increased absorption of incident light due to the red, visibly opaque water occurred during our STT, and may have enhanced the STT magnitude, leading to an inflated temperature trigger and peak eggs prediction. The summer fish eggs, on the other hand, would be expected to be substantially reduced by the spring fish kills caused by the red tide.

Despite this potential source of error external to our model, any failure of an out-of-sample prediction deserves careful attention (indeed, the prediction from Ref. [15] still holds in 2020). The fact that the peak observed fish egg count in 2020 actually did not occur during the summer (with a substantially larger peak in the spring which again may be related to the red tide; Fig 4A), prompted us to question whether the peak eggs should be defined as a summer event, or as an annual event. Relaxing this timing definition, and instead looking for an annual peak in eggs with a triggering event that occurs at *any time* before, completely disrupts the temperature trigger to peak eggs relationship, and not just for the 2020 data (Fig 4B; see Methods). This suggests the possibility of overfitting in the base definitions, that is confirmed by allowing the date defining the boundary between spring and summer to vary (Fig 4C). We find that

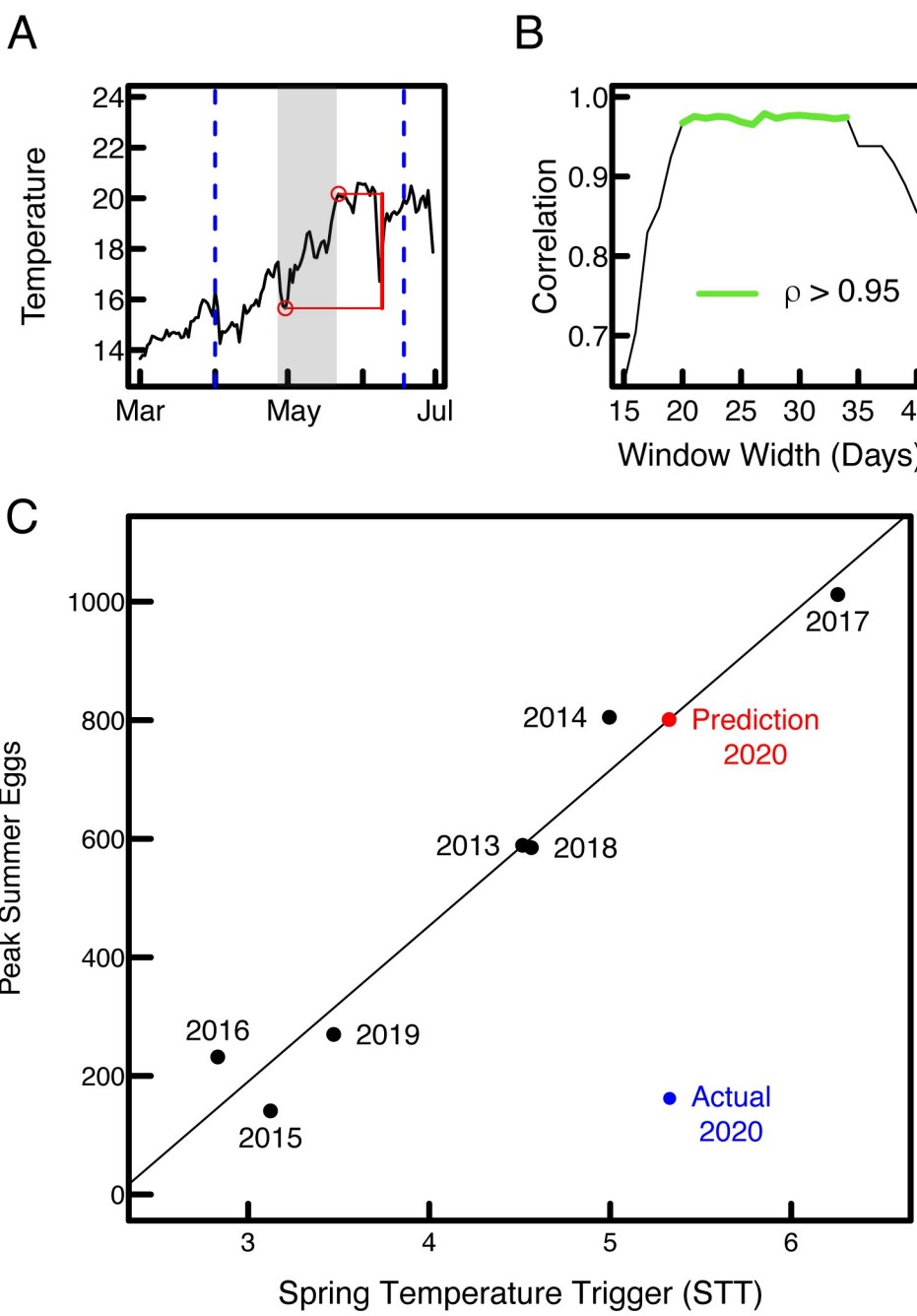

**Fig 2. An apparently robust temperature trigger for peak summer egg abundance that ultimately failed in true out-of-sample prediction.** A) We defined the spring temperature trigger (STT) as the largest temperature increase (denoted in red) detected within a monthly sliding window (gray area) as it moves in daily increments over the spring season (dashed lines; see Methods). B) The relationship between STT and peak summer eggs was robust to the width of the sliding window (widths that produce a ρ > 0.95 for the data up to and including 2019 are indicated in green). C) The peak correlation between STT and peak summer eggs (June–August) for 2013–2019 (black dots) and predicted value of 801 eggs for 2020 based on the linear regression (red dot), which differs dramatically from the eventually observed peak summer eggs (blue dot).

only small variations in the boundary date between spring and summer can substantially reduce the observed correlation (Fig 4C) which strongly suggests that the original observed relationship may have been a case of over-fitting.

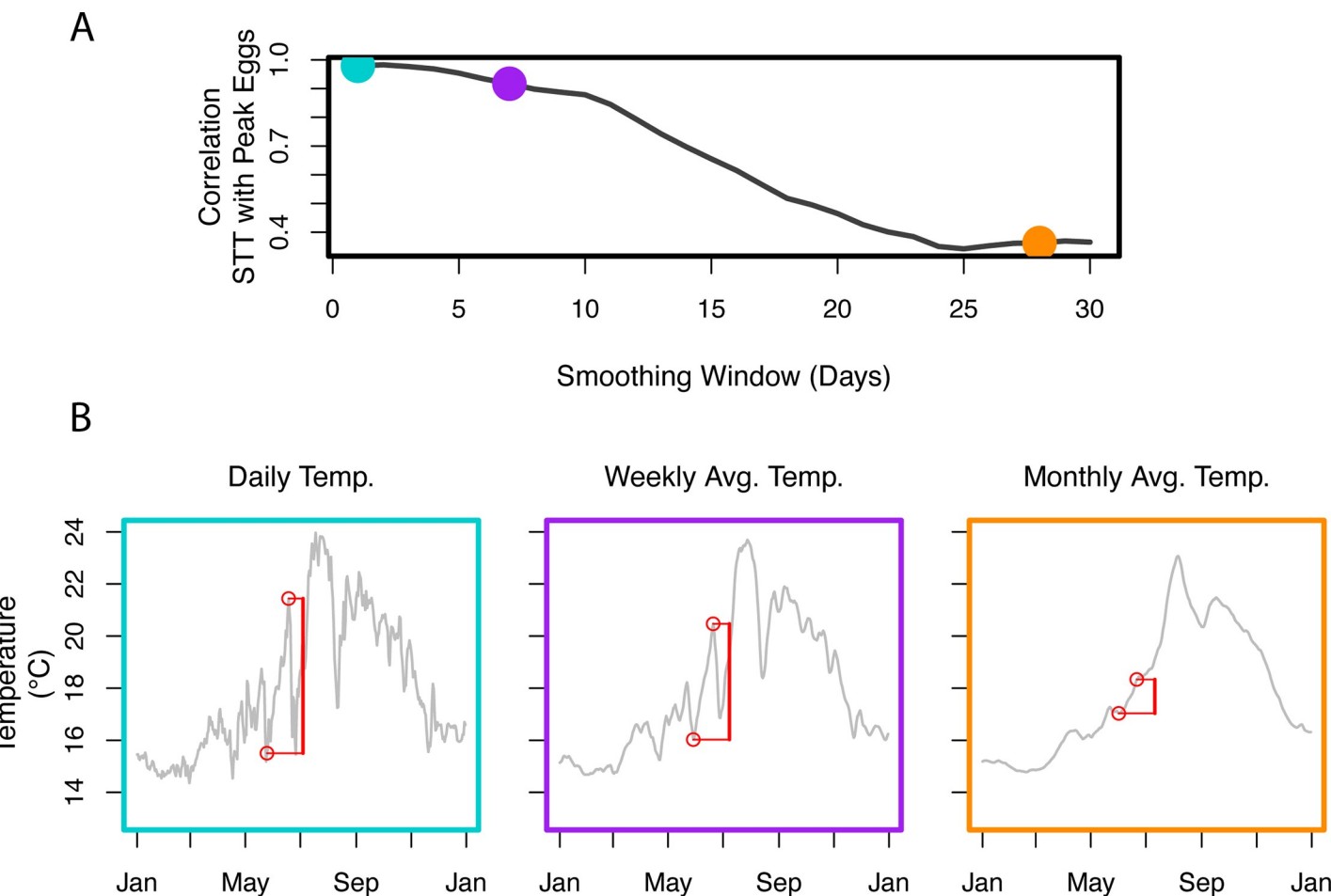

**Fig 3. Time averaging of the 2013–2019 data obscured the original spring temperature trigger, suggesting the predictive information lay in the daily-resolution temperature data, not in the trend.** A) The relationship between spring temperature trigger (STT defined over 27 days) and maximum summer fish egg abundance declined as SST was increasingly smoothed across the x-axis (from daily to monthly averages). B) The daily (blue), weekly averaged (purple), and monthly averaged (orange) sea-surface temperature in 2017. Note how the magnitude of the STT (red bars) declines with averaging. Note that this figure does not include 2020 data.

## Discussion

Many ecological models and relationships are based on a large number of parameters, which renders the task of assessing model robustness in that high-dimensional parameter space generally difficult. However, even more difficult to assess is the robustness of assumptions that underlie the construction of the model itself, from which the model parameters arise. These assumptions may be non-quantitative (e.g. categorical), such as the choice of functional form or the inclusion or exclusion of model elements. Devising appropriate all-encompassing tests across this full space of possible hypotheses is neither feasible nor advisable, and so this source of uncertainty is often overlooked. Often, however, some assessment of the structural stability of models is both possible and highly enlightening (e.g. [31]) and should be encouraged. In the case of the STT to peak summer eggs relationship discussed here, the underlying definition of the spring and summer intervals was quantitative, and so was particularly easy to assess, but it is often the case with assumptions that they seem so obvious as to become invisible.

The process of making true out-of-sample predictions can account for both the difficulty of seeing potential sources of overfitting, and the difficulty of assessing them. Furthermore,

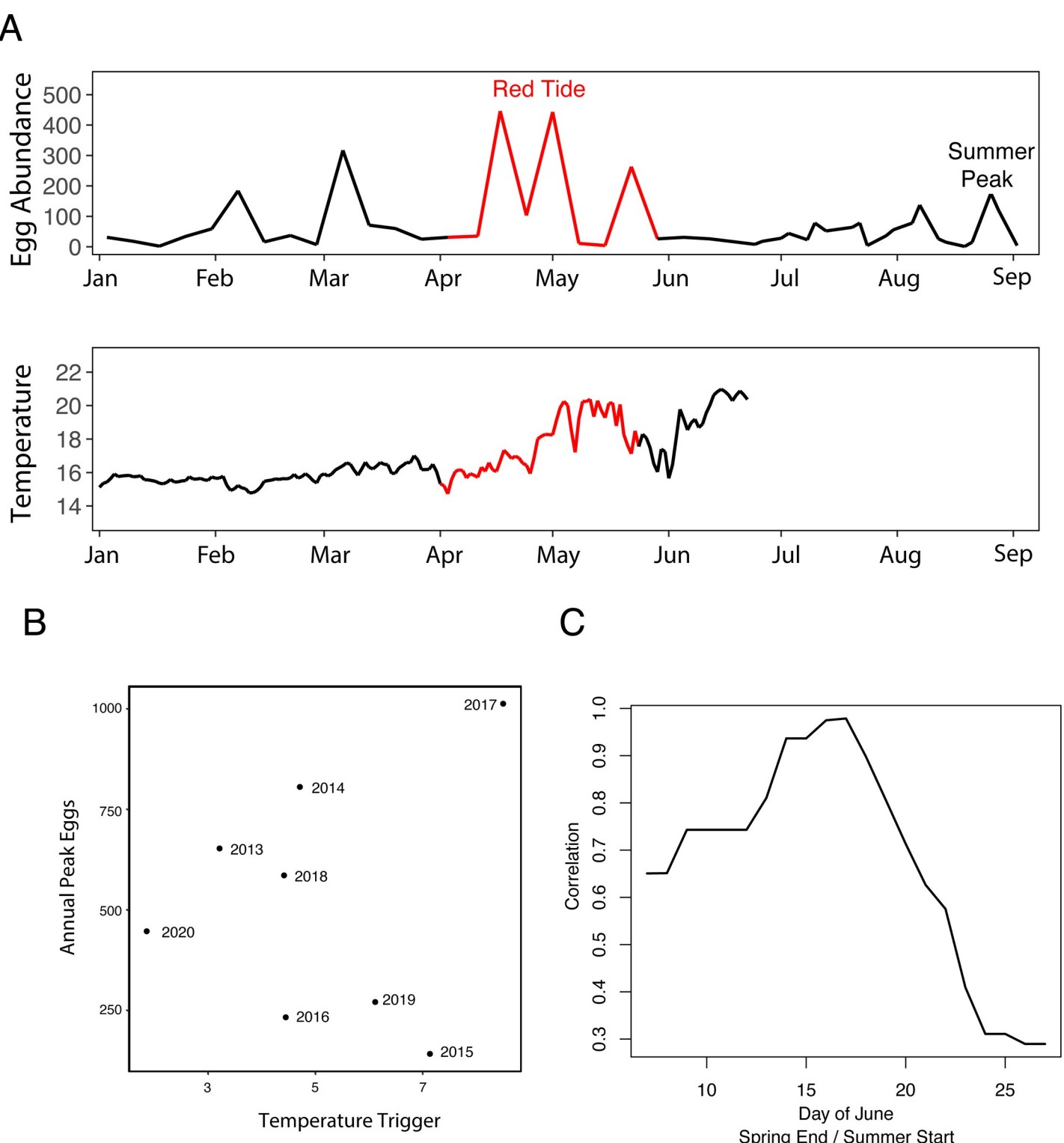

**Fig 4. The failure of our 2020 prediction and the structural sensitivity of the model.** A) 2020 fish eggs and temperature time series showing the timing of the red tide. The timing of the red tide was estimated based on the duration of anomalous dissolved oxygen levels measured by the Scripps Ocean Acidification Real-time (SOAR) Monitoring Program. B) Correlation between a triggering event that occurs any time before the annual peak eggs, and the annual peak eggs from 2013 to 2020 suggests possible overfitting (see Methods). C) Sensitivity of the original 2013–2019 STT to peak summer eggs relationship to variation in the spring end date and summer start date further suggests overfitting.

publishing predictions before it is possible to assess their accuracy places in clear view any subsequent revisions of the model or relationship to account for new data. This may seem to go against natural intuition: if a model prediction can be tested, should it not be tested before publication, to improve confidence in the model? Certainly, a confidence threshold must be reached in order to justify publication (as it was here, where because of the nearly perfect fit any subsample with 3 or more points would reasonably produce successful out-of-sample predictions). However, we suggest that once that confidence threshold has been reached, leaving a true out-of-sample validation avenue open to future investigation is a valuable standard to aspire to, and we support the push towards quantitative prediction of truly out-of-sample natural data as a validation standard for ecological and earth systems science.

## Materials and methods

### Convergent cross mapping

Convergent cross mapping was performed using the block_lnlp() function in rEDM *v*0.7.3 [32]. The embedding was made with the raw egg abundance data and the daily-averaged SCCOOS temperature time series.

When performing CCM, the variable being predicted is the one being tested as a causal driver. As such, we used the egg abundance time series to predict temperature (thus measuring temperature's effect on egg abundance). Although temperature data existed for every day, eggs were collected at inconsistent intervals, typically ranging from one collection every 2–5 days. Thus, in order to make a proper embedding, we filtered both temperature and egg abundance time series to only include temperature values for which a collection occurred on a given day, 6–8 days prior, and 13–15 days prior as well. This gave us a 3-dimensional embedding for fish eggs, with time lags of about 1 week, with accompanying temperature values.

Because both egg abundance and temperature are strongly seasonally driven, we needed to make sure we were not identifying shared information in the two variables driven by seasonality. To account for this, nearest neighbor selection only considered time points that were within 90 calendar days for our target prediction. Without doing this, increased library size will only increase the amount of seasonal information resolved in the embedding rather than actual causal inference.

Libraries of potential neighbors (points within 90 calendar days of the date of the target) were generated at random for each predicted point. Library sizes ranged from 10–80 points (increasing by increments of 5). Once the library was randomly generated, the nearest 4 neighbors (E+1, see (22)) in state space were selected and used to make a prediction. After a prediction was made on each temperature value, Pearson's correlation was calculated between observed and predicted values. This process was repeated 50 times for each library size.

### STT calculation

As described in the text and illustrated in Fig 2, STT was calculated by allowing the last day of the sliding window to move over the spring interval defined as April 1st to June 18th.

### Sensitivity test

The annual peak in egg abundance was identified as the maximum egg abundance occurring between January 1st and December 31st within each year. The trigger-like value was found using the sliding window analysis described in the text and illustrated in Fig 2, however the last day of the sliding window was allowed to move from January 28th to the day immediately preceding the annual peak in eggs.

## Sample collection

California Department of Fish and Wildlife permit (#4564) was used for the collection of plankton from the MPA's. Vertical plankton tows (approximately weekly) were conducted off of the Scripps Pier (32.8328˚ N, -117.2713˚ W) from 2013 to 2019. A 1-meter diameter net with 505 micron mesh and a bottle attached to the cod end was lowered to the seafloor, approximately 5 meters, and out of the water 4 times, sampling a total of ~16 cubic meters of seawater. The net was then rinsed by lowering it into the water until the top of the net touched the surface and then raised back out. It is worth noting that this method only samples eggs suspended in the water column and does not effectively collect demersal eggs. There is some variation in the volume of water being sampled (e.g., water depth changes with tide), but since all collections went from seafloor to surface we do not expect any significant effect of egg depth profiles to have any significant effect on our measurement peak summer egg abundance. Currents could also affect sample volume but are rarely strong in the summer [33] and are therefore less likely to skew the value of peak eggs. The eggs captured at the pier all originated 0–3 days before the collection occurred since in Southern California water temperatures, most eggs for the fish species found there hatch within 72 hours [34] hence any eggs from a spawning event preceding the collection by up to 3 days could still be represented in our sample, depending on precise spawning location and currents. Using real-time current velocities, retrospective modeling found that most eggs collected at the Pier site likely originated within a few kilometers of the collection site [34]. The contents of the cod end were concentrated through a 330 micron mesh screen and then sorted under a microscope at 10X. The fish eggs were counted, placed in 1.5 mL tubes containing 95% ethanol, and stored at -20˚C for at least 24 hours until further processing. At this step, the morphologically distinct eggs of the Northern anchovy (*Engraulis mordax*) and Pacific sardine (*Sardinops sagax*) are counted and stored separately because they do not require molecular methods for identification. The remaining eggs are identified through DNA barcoding. Comprehensive species lists are found in S2 and S3 Tables.

## Supporting information

**S1 Fig. Correlation between multiple variables studied.** The strongest correlation we found was between STT and peak egg abundance (maximum correlation of 0.98). However, as found by [15], a strong, negative correlation exists between average winter temperatures and average summer egg abundance. Not surprisingly, there is also a strong correlation between peak egg abundance and average egg abundance for a given summer (A). Due to transitivity, there is also a strong correlation between average summer eggs and spring temperature triggers (B). Weaker correlations also exist between average winter temperatures and the finer scale temperature triggers (C) and peak summer egg abundance (D), however these are much weaker relationships ($p > 0.1$).
(TIF)

**S2 Fig. Inconclusive evidence of between-species synchrony in peak summer egg abundance.** Shannon diversity (base *e*) appears to be higher for peaks with lower abundance.
(TIF)

**S1 Table. Species composition of the peak summer egg abundance samples.** The proportional contribution that each of the identified species contributes to the annual peak summer egg abundance. The peak samples in each year are dominated by the eggs of a few species, with the dominant species varying from year-to-year.
(DOCX)

**S2 Table. Scripps pier species abundance 2013–2019.** A list of all 46 species identified in the samples from Scripps Pier from 2013–2019 and the number of eggs identified as each of those species within each year. The sampling effort by year is as follows: 2013 = 161, 2014 = 84, 2015 = 51, 2016 = 52, 2017 = 48, 2018 = 75, 2019 = 65.
(DOCX)

**S3 Table. Scripps pier species frequency 2013–2019.** A list of all 46 species identified in the samples from Scripps Pier from 2013–2019 and the proportion of samples they were present in within each year. The sampling effort by year is as follows: 2013 = 161, 2014 = 84, 2015 = 51, 2016 = 52, 2017 = 48, 2018 = 75, 2019 = 65.
(DOCX)

## Author Contributions

**Conceptualization:** Emma S. Choi, Erik Saberski, Tom Lorimer, George Sugihara.

**Data curation:** Emma S. Choi, Ronald S. Burton.

**Formal analysis:** Erik Saberski, Tom Lorimer, Cameron Smith.

**Funding acquisition:** Ronald S. Burton, George Sugihara.

**Investigation:** Emma S. Choi, Erik Saberski, Tom Lorimer, Unduwap Kandage-don, George Sugihara.

**Methodology:** Erik Saberski, Tom Lorimer, Cameron Smith, George Sugihara.

**Project administration:** George Sugihara.

**Resources:** Ronald S. Burton.

**Supervision:** Tom Lorimer, Ronald S. Burton, George Sugihara.

**Validation:** Emma S. Choi, Cameron Smith.

**Writing – original draft:** Emma S. Choi, Erik Saberski, Tom Lorimer, George Sugihara.

**Writing – review & editing:** Emma S. Choi, Erik Saberski, Tom Lorimer, Unduwap Kandage-don, Ronald S. Burton, George Sugihara.

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
