## [Decision Letter · Decision Letter 0]

6 Aug 2020

PONE-D-20-21189

Temperature triggers provide quantitative predictions of multi-species fish spawning peaks

PLOS ONE

Dear Dr. Sugihara,

Thank you for submitting your manuscript to PLOS ONE. After careful consideration, we feel that it has merit but does not fully meet PLOS ONE’s publication criteria as it currently stands. Therefore, we invite you to submit a revised version of the manuscript that addresses the points raised during the review process.

In your revision please address all the comments and suggestions made by reviewer #1. In particular also make sure that the data are freely available and that the link provided to access the data works, thank you.

We look forward to receiving your revised manuscript.

Kind regards,

Andrea Belgrano, Ph.D.

Academic Editor

PLOS ONE

Journal Requirements:

Reviewers' comments:

Reviewer's Responses to Questions

**Comments to the Author**

1. Is the manuscript technically sound, and do the data support the conclusions?

Reviewer #1: Yes

2. Has the statistical analysis been performed appropriately and rigorously? 

Reviewer #1: Yes

3. Have the authors made all data underlying the findings in their manuscript fully available?

Reviewer #1: No

4. Is the manuscript presented in an intelligible fashion and written in standard English?

Reviewer #1: Yes

5. Review Comments to the Author

Reviewer #1: This is an interesting analysis. It was a pleasure to read it. I favor publication. I am left with few questions, and a suggestion, which I hope the authors can address in a review.

1. What is the species composition of the egg catch? I am unsure with the statement that eggs are buoyant. Many are not, and it depends on the species.

2. Somewhat related to my previous question, Fig S2 is interesting and it shows decline of egg diversity in relation to peak eggs abundance. This is presumably driven by an increase of dominance of few or a single species. Can you elaborate on the species that dominate the samples, especially when egg abundance increases? At the end of the methods the authors indicate that the morphologically distinct anchovy and sardine eggs were removed, and the rest of the eggs were counted and identified to species using DNA barcoding. Would be great to see the species list.

5. This is a comment/suggestion. I suggest redirecting the focus of the research question toward a more mechanistic relationship between temperature and egg abundance. The author ask whether 'finer-timescale temperature dynamics provide information about finer-timescale fish egg abundance dynamics.' However, the striking relationships that they have uncovered between STT and peak egg abundance, in my view, is still an integrated measure rather than an examination of a finer scale relationship between temperature and eggs. There is still value in this relationship of course, but not of the same type suggested by the author. This analyses reveals potential mechanisms, pointing to the fact that large variations of water temperature during spring, may trigger massive spawning events during summer.

6. PLOS authors have the option to publish the peer review history of their article (what does this mean?). If published, this will include your full peer review and any attached files.

Reviewer #1: **Yes: **Lorenzo Ciannelli

---

## [Author Response · Author response to Decision Letter 0]

22 Oct 2020

Response to Reveiwer 1

Reviewer’s comment: What is the species composition of the egg catch? I am unsure with the statement that eggs are buoyant. Many are not, and it depends on the species.

The Reviewer raises an important point, and we’re sure many people will be interested in the composition of the samples. To address this, we have included two new supplemental tables. The first of these two tables, Table S2: Scripps Pier Species Abundance 2013–2019, lists all 46 of the species identified from the sampling at Scripps Pier from 2013 to 2019 and the number of eggs identified as each of these species within each year. The second of these two tables, Table S3: Scripps Pier Species Frequency 2013–2019, lists all 46 of the species and the proportion of samples (out of the yearly sampling effort) in which eggs from the species were observed within each year.

With regard to buoyancy, we have added a statement to the manuscript that our methods only effectively sample eggs suspended in the water column; we found very few eggs from species with demersal eggs – those were presumably stirred off the bottom by the net.

Reviewer’s comment: Somewhat related to my previous question, Fig S2 is interesting and it shows decline of egg diversity in relation to peak eggs abundance. This is presumably driven by an increase of dominance of few or a single species. Can you elaborate on the species that dominate the samples, especially when egg abundance increases? At the end of the methods the authors indicate that the morphologically distinct anchovy and sardine eggs were removed, and the rest of the eggs were counted and identified to species using DNA barcoding. Would be great to see the species list.

We thank the Reviewer for highlighting this opportunity for clarification. We have now included a new supplemental table, Table S1: Species composition of the peak summer egg abundance samples, that lists the proportion of the annual summer peak eggs that were identified as each species. This table highlights which species dominate the peak summer egg abundance samples. We have slightly expanded the main text in the discussion of synchrony to accommodate the addition of this table. 

Reviewer’s comment: This is a comment/suggestion. I suggest redirecting the focus of the research question toward a more mechanistic relationship between temperature and egg abundance. The author ask whether 'finer-timescale temperature dynamics provide information about finer-timescale fish egg abundance dynamics.' However, the striking relationships that they have uncovered between STT and peak egg abundance, in my view, is still an integrated measure rather than an examination of a finer scale relationship between temperature and eggs. There is still value in this relationship of course, but not of the same type suggested by the author. This analyses reveals potential mechanisms, pointing to the fact that large variations of water temperature during spring, may trigger massive spawning events during summer.

While we greatly appreciate this suggestion, and would like to be able to pursue it, it is not currently possible to redirect our research question to address the finely resolved details of the mechanistic relationship between temperature and egg abundance. This is because we did not specifically measure a variable to demonstrate how temperature is acting to influence fish reproduction. Given that we are measuring a macroscopic output variable, egg abundance, it is difficult to determine whether the STT is having a direct physiological effect on the fish or whether it is related to other more proximate factors that drive increases in egg abundance. These are questions perhaps best answered by experimental manipulation; in the current experimental design of our study we could not discern the finer-scale details of the mechanism at play. The causality detection method (CCM), however, does verify (within these data) that there is a causal link here between temperature and egg abundance insofar as changes in temperature propagate to changes in egg abundance. Moreover, we show that the strength of the relationship identified here is dependent upon fine time scale measurements – both the STT and the peak summer egg abundance are captured through frequent measurements, daily in the case of STT and weekly in the case of peak summer egg abundance. In Figure 3A we showed that in smoothing the daily temperature datum, we lose the signal between the STT and peak summer egg abundance. Therefore, we focused our research question on fine time scale dynamics that are shown to be essential to this analysis, rather than a mechanistic relationship that we are unable to speak to, given the output variable we measured.

---

## [Editor Report · Decision Letter 1]

2 Nov 2020

The importance of making testable predictions: a cautionary tale

PONE-D-20-21189R1

Dear Dr. Sugihara,

We’re pleased to inform you that your manuscript has been judged scientifically suitable for publication and will be formally accepted for publication once it meets all outstanding technical requirements.

Kind regards,

Andrea Belgrano, Ph.D.

Academic Editor

PLOS ONE

Additional Editor Comments (optional):

The revised manuscript addresses All the comments/suggestions made by reviewer #1 including the availability/access to All the data that you have now nicely provided via the GitHub repository. The manuscript makes an interesting and transparent "tale" that I believe will be of great interest to the broad readership of PLOS ONE.

---

## [Editor Report · Acceptance letter]

26 Nov 2020

PONE-D-20-21189R1 

The importance of making testable predictions: a cautionary tale 

Dear Dr. Sugihara:

I'm pleased to inform you that your manuscript has been deemed suitable for publication in PLOS ONE. Congratulations! Your manuscript is now with our production department. 

Kind regards, 

on behalf of

Dr. Andrea Belgrano 

Academic Editor

PLOS ONE